# Identifying and Interpreting Non-Aligned Human Conceptual Representations using Language Modeling

**Wanqian Bao & Uri Hasson**
Center for Mind/Brain Sciences (CIMeC)
University of Trento
Rovereto, 38068, Italy
`wanqbao8596@gmail.com, uri.hasson@unitn.it`

## Abstract

The question of whether people's experience in the world shapes conceptual representation and lexical semantics is longstanding. Word-association, feature-listing and similarity rating tasks are methods that aim to address this question but ultimately require a subjective interpretation of the latent dimensions or clusters identified. In this study, we introduce a supervised representational-alignment method that (*i*) determines whether two groups of individuals share the same basis of a certain category, and (*ii*) explains in what respects they differ. In applying this method, we show that congenital blindness induces conceptual reorganization in both a-modal and sensory-related verbal domains, and we identify the associated semantic shifts. We first apply supervised feature-pruning to a language model (GloVe) to optimize prediction accuracy of human similarity judgments from word embeddings. Pruning identifies one subset of retained GloVe features that optimizes prediction of judgments made by sighted individuals and another subset that optimizes judgments made by blind. A linear probing analysis then interprets the latent semantics of these feature-subsets by learning a mapping from the retained GloVe features to 65 interpretable semantic dimensions. We applied this approach to seven semantic domains, including verbs related to motion, sight, touch, and amodal verbs related to knowledge acquisition. We find that blind individuals more strongly associate social and cognitive meanings to verbs related to motion or those communicating non-speech vocal utterances (e.g., whimper, moan). Conversely, for amodal verbs, they demonstrate much sparser information. Finally, for some verbs, representations of blind and sighted are highly similar. The study presents a formal approach for studying interindividual differences in word meaning, and the first demonstration of how blindness impacts conceptual representation of everyday verbs.

## 1 Introduction

### 1.1 Conceptual representation in congenitally blind

The question of whether words capture similar meanings across different languages or among different speakers of the same language is central to understanding the relation between language and conceptual representation. Thompson et al. (2020) used a computational modeling approach to show that representations of non-specialized, everyday words can substantially differ across cultures. The question also strongly applies to speakers of the same language. Acquiring sub-cultural lexicons (e.g., slang terms) clearly requires learning and is associated with changes in meaning over time (e.g., Friendly & Glucksberg, 1970). But does one's personal experience produce different meanings for everyday words such as 'touch,' 'see,' and 'learn'? One approach to tackling this issue has been to study the nature of word meaning in blind individuals, directly evaluating whether sensory experiences in the world can impact knowledge of word-meaning. Indeed, this question has interested cognitive psychologists for decades (for recent reviews, see Mamus et al., 2023; Bedny et al.,

2019). This work has produced substantial evidence that the blind use language in a way indicative of having knowledge of the visual world, and of the use of vision-related words.

However, the question of how blind and sighted differ in conceptual organization is still open. Multiple approaches have been used to quantify semantic knowledge in the blind, including the production of definitions (e.g., Landau & Gleitman, 1985), word-associations (e.g., Lenci et al., 2013), event-description (Mamus et al., 2023) and similarity judgments for colors (e.g., Shepard & Cooper, 1992), among others. More recently, Bedny et al. (2019) obtained similarity judgments for different types of verbs, and examined the data using representational similarity analysis and other descriptive techniques. An important finding was that for some verb types, there was a resemblance in similarity judgments between the two groups. In particular, verbs related to visual stimulation (e.g., 'sparkle', 'blink') were judged in a consistent manner. Interestingly, Lewis et al. (2019) subsequently showed that these similarity judgments were well predicted by (proxies for) similarity values produced from a language model (word2vec; Mikolov et al., 2013). The authors suggested that blind and sighted learn similar representations because of similar exposure to language. These prior studies however could not formally and precisely explicate how the conceptual representations of blind and sighted differ when representing everyday words.

## 1.2 CURRENT APPROACH: SUPERVISED LEARNING OF REPRESENTATION

The current study aims to determine whether blind and sighted individuals use similar information when representing verb meanings. To address this, we adopt a two-step computational modeling approach, each verified using cross-validation. In step 1 we use supervised-pruning to fine-tune a language model to predict human similarity judgments. In step 2, we examine the information latent in the fine-tuned model's feature space, studying both the retained vectors (word embeddings) and the information contained in them, using a probing task. In sum, we investigate both the basis vectors and the representations formed in (subspaces of) language-models fine-tuned to predict blind or sighted behavior.

We present an informal description of the logic here. Details are provided in Methods, and a more formal algorithmic presentation is given in the Appendix. The workflow is summarized in Figure 1, and we refer to elements within that figure here. To begin, we first prune GloVe word embeddings (*1*) to improve prediction of human similarity judgments (*2*). This produces an embedding matrix based on a smaller set of retained features (*3*). This is done separately for judgments provided by congenitally blind and sighted controls, across seven different categories of verb types (anticipating the results, these produced divergent retained feature sets for blind and sighted). After pruning, we examine information in the retained feature sets using a probing task (Belinkov, 2022). The probing task is based on a Partial Least Squares Regression (PLSR) function, and takes as its target a dataset provided by Binder et al. (2016), which includes 535 words rated on 65 dimensions (*5*). It has been shown (Chersoni et al., 2021) that it is possible to predict the human ratings for these 535 words from their GloVe features, using PLSR. We instead learn a mapping between the the embeddings of these 535 words using only the retained features (*4*) and the 65 human annotated dimensions (*5*), which produces predictions of the 65 dimensions for each word (*6*). This step is performed separately using the features retained for blind and sighted. Crucially, by comparing the predicted values (*6*) to the ground-truth values (*5*), the probing task allows quantifying the relative salience of 65 semantic

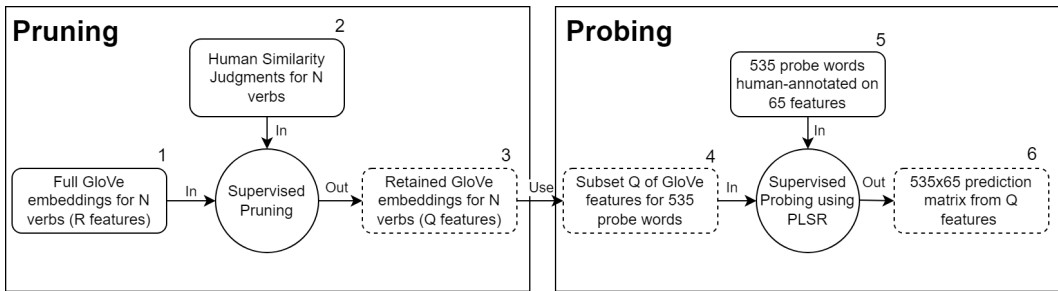

Figure 1: Analysis workflow consisting of supervised pruning and supervised probing. Solid lines indicate input datasets and dashed lines indicate products of analyses

dimensions in the retained feature sets used in the mapping. In addition, using the probing task we can also determine which of the 65 Binder features are differentially predicted for blind and sighted. We use pruning and probing after showing they perform effectively in cross-validation.

## 2 METHOD

### 2.1 DATASETS AND WORD EMBEDDINGS

For similarity judgments we used data made publicly available by Bedny et al. (2019). Twenty-five congenitally blind and 22 sighted individuals rated semantic similarity of verb-pairs on a scale of 1 to 7 (7 being most similar). The materials were 119 verbs divided into seven categories (we follow the original terminology). Each category contained 15 verbs apart from one category that contained 14. These verb categories were: Perception Sight (e.g., see, look), Perception Touch (e.g., pat, rub), Perception Amodal (e.g., learn, discover), Emission of Light (e.g., twinkle, glow), Emission of Animate Sounds (e.g., groan, growl), Emission of Inanimate Sounds (e.g., buzz, beep), and manner of Motion verbs (e.g., float, bounce). A complete list is given in the Appendix. Similarity ratings were normalized within participants to create similarity matrices, with matrices calculated separately for each verb category and averaged for the group. Furthermore, in our analysis we excluded three participants who did not complete the rating task. This ultimately produced data for 24 blind and 20 sighted participants.

For the probing task we used semantic annotations collected by Binder et al. (2016). Using an online survey, they obtained human feature-ratings for 535 words. Participants indicated the extent to which each word was linked to 65 distinct semantic attributes, covering meanings across 14 overarching domains. For certain words, some attributes were inapplicable. For instance, the dimensions of time and duration were not applicable to words like 'shoe'. In these cases participants indicated zero. The original dataset contained 535 words by 65 dimensions from 14 categories, However, the word 'used' appeared twice: once as verb and once as adjective. In our analyses, the results did not change whether one or the other were used and so we deleted one and analyzed the remaining 534 words.

For word embeddings we used GloVe (Pennington et al., 2014) as its embeddings perform well in predicting word similarity judgments (e.g., Richie & Bhatia, 2021). We used the 300-dimensional GloVe vectors sourced from the Wikipedia 2014 + Gigaword 5 corpus, which contains approximately 400,000 uncased vocabulary entries, with a total 6 billion tokens. We extracted the embeddings of those words that were used in Bedny et al. (2019) and Binder et al. (2016).

### 2.2 SUPERVISED PRUNING

#### 2.2.1 PRUNING ALGORITHM

We used a supervised pruning algorithm, which is a wrapper-based sequential feature selection procedure optimizing the alignment between a human similarity matrix and one generated from an embedding space. Its objective is to identify a reduced subset of features, so that when that subset is used to generate the $Word \times Word$ similarity matrix, the resulting matrix produces a maximal fit to the human similarity judgments. Pairwise similarity between words was computed from word embeddings using cosine similarity, while the fit between the two similarity matrices was computed using the Spearman's Rho rank correlation coefficient ($\rho$), following Richie & Bhatia (2021). Pruning was applied in two ways. In the first approach, pruning was applied by considering the entire set of similarity judgments associated with each verb category. This identifies the subset of features that optimizes prediction of similarity judgments for the verbs in that category. For instance, if a category consisted of 15 verbs, there were 105 unique similarity judgments that supervised the pruning procedure. The second approach used cross-validation to demonstrate that pruning indeed identifies meaningful dimensions, as indicated by out-of-sample generalization. In each fold, one word was designated as the target, and the test set only included those similarity judgments involving the target word. The training set consisted of all other verb-pairs within the category. For example, in a verb category with 15 words, the test set comprised the 14 word-pairs where the target word appeared. Baseline performance for the test set was established by determining the prediction accuracy for matched human data using the complete set of 300 GloVe features. This baseline was compared to performance when using only the features retained from the training set. As an

additional control, in each case we also constructed a random feature-set, matched in size to that of the retained set, and used that for the test-set. Both approaches were applied to similarity matrices associated with each of the seven verb types in Bedny et al. (2019), as produced by blind and sighted (i.e., 14 different times in total).

### 2.2.2 OVERLAP BETWEEN FEATURE SETS RETAINED BY PRUNING

Because pruning identifies a subset of features, it is possible, for any two results of the pruning algorithm, to compare the similarity between the two sets of retained features. This computation operates on the indices of the feature sets retained via pruning. As an indicator of association we used the Dice coefficient of two sets $U, V$: Dice(U,V) = $(2 * |U \cap V|)/(|U| + |V|)$. Given that we compared sets that often varied in size, in such instances, we adjusted the larger set to match the size of the smaller set, by selecting the $n$ top-ranking features from the larger set where $n$ is the size of the smaller set. This assured that both sets contributed equally to the denominator term.

It is important to keep in mind that two distinct semantic domains (e.g., domains A and B) with ostensibly different meanings (i.e., low similarity between words across the two domains) can still converge on the exact same feature sets through pruning. This can occur as long as the retained features consistently have different values for words in domains A and B. In other words, pruning identifies features that form a basis set for characterizing the structure of a domain, while the similarity ratings are determined by the values on those features across objects. To illustrate, consider words in two domains, A and B (e.g., two types of nouns), where pruning retains only features a, b, c, d for both domains. For words in domain A, the values of (a,b) are consistently higher than (c,d), with the remaining variance accounting for similarity within the domain. Conversely, for words in domain B, the values of (a,b) are consistently lower than (c,d). Although the similarity of objects within A and within B is greater than the similarity across domains, both domains rely on the exact same features. Thus, whether the same features are required to account for the similarity structure of two domains is independent of the domains' similarity.

## 2.3 SUPERVISED PROBING

Following previous work (e.g., Chersoni et al., 2021; Utsumi, 2020; Flechas Manrique et al., 2023), we used Partial Least Squares Regression (PLSR) to predict the 65 Binder dimensions for a dataset of 535 words for which we also extracted GloVe embeddings. We follow the exact same procedure, applying it separately to each of the 14 retained feature sets (seven verb types for blind/sighted) found via supervised pruning. We emphasize that for each verb category, we only use the GloVe features that were identified for that category in the pruning step (i.e., we do not use all 300 features). The outcome of this prediction allows determining if sets retained by different domains encode different semantics. The retained features used for PLSR were produced taking into account the entire set of human similarity judgements (i.e., outside a cross-validation context).

We applied PLSR to the retained sets, employing a leave-one-out cross-validation procedure (following Chersoni et al., 2021). Binder's dataset included 535 words, with 534 remaining after removing the dual use of the word 'use'. For each fold, we designated 533 words as the training set and the single left-out word as the test set. PLSR was trained on the training set, and the model was then applied to the test word, predicting its 65 feature values. Note that when applied to all 534 words this produces a $534 \times 65$ stacked matrix, including all words, which we refer to as a leave-one-out-cross-validation (LOOCV) Stacked Matrix 65; $LSM_{65}$. The stacked matrix was produced by 534 different regressions. We further note that in some analyses we condense the $534 \times 65$ matrix into $534 \times 14$ by merging the 65 dimensions into the 14 broader semantic areas, producing $LSM_{14}$. The merging of the 65 features to 14 domains was implemented according to an assignment specified in Binder et al. (2016) (see their Tables 1, 2).

The data in the prediction matrices serve as the basis for evaluating differences in the semantic content of embeddings pruned for blind and sighted, and for different verb types. We perform three analyses: First, considering $LSM_{65}$ by column (feature), we correlated the predicted values of each column ($n = 534$) with Binder's ground truth data. This is an indicator for the accuracy by which each feature was predicted, used by Chersoni et al. (2021). These data were were also averaged into the 14 larger semantic domains. Second, considering $LSM_{65}$ by row (word), we correlated the predicted values of each row with the ground truth data. Third, to determine whether there were

| | CB Base | CB Retained | CB F. | S Base | S Retained | S F. |
|---|---|---|---|---|---|---|
| Perc. Amodal | 0.41 | **0.39** {0.32} | 74 | 0.47 | **0.46** {0.37} | 84 |
| Perc. Sight | 0.56 | **0.63** {0.48} | 118 | 0.64 | **0.68** {0.50} | 113 |
| Perc. Touch | 0.16 | **0.40** {0.19} | 76 | 0.14 | **0.38** {0.08} | 58 |
| Emission. Light | 0.56 | **0.62** {0.54} | 99 | 0.42 | **0.54** {0.41} | 80 |
| Emission. Animate Sound | 0.25 | **0.26** {0.27} | 67 | 0.33 | **0.37** {0.28} | 59 |
| Emission. Inanimate Sound | 0.10 | 0.09 {0.08} | 46 | 0.13 | 0.06 {0.07} | 57 |
| Motion | 0.35 | 0.29 {0.22} | 75 | 0.33 | **0.40** {0.18} | 60 |

Table 1: Out of sample generalization of supervised pruning. Base (Baseline) indicates prediction when using all 300 GloVe features. Retained indicates prediction using features learned from training set. CB = Congenitally Blind ; S = Sighted. F. (Features) indicates average number of features retained from training set. Numbers in curly brackets in the 'Retained' columns define chance and indicate prediction performance when using randomly selected feature sets matched in size to those retained from the training set. Values in bold indicate better prediction than baseline, or similar prediction to baseline that surpasses the chance value.

differences in the prediction-accuracy from pruned embeddings from blind and sighted, we also defined accuracy as the absolute deviation between the predicted value and ground-truth value for each cell in $LSM_{65}$. Taking words ($N = 534$) as unit of analysis, these cell-wise prediction errors allow comparing prediction accuracy, for each of the 14 domains, for blind and sighted. We use paired T-tests, with words as the unit of analysis.

## 3 RESULTS

### 3.1 PRELIMINARY EVALUATION: OUT OF SAMPLE GENERALIZATION OF PRUNING SOLUTION

Table 1 shows prediction for test-set data when using all GloVe features (baseline) or those learned via pruning of the training set ('retained'). Pruning often generalized successfully: the subsets of features learned via pruning either exceeded baseline performance, or maintained baseline performance using only between 20-30% of the total 300 features. In three cases we did not find generalization: for Emission-InanimateSounds (sighted and blind) and for Emission-AnimateSounds (for blind). In the latter case the performance from full set, retained set and random set were all very similar ($\approx 0.27$). Note that all data reported in Table 1 are based on prediction-performance computed for test-folds partitions consisting of 14 data points, which may limit sensitivity. In the following analysis we apply and evaluate pruning on the same, complete dataset.

### 3.2 CORRESPONDENCE BETWEEN FEATURE-SETS RETAINED BY PRUNING

Table 2 presents the outcomes of the pruning process applied to individual verb datasets, conducted outside the context of a cross-validation framework. While this may be treated as over-fitting from a machine learning perspective, our aim here is to obtain the most effective predictor of a specific set of human similarity judgments. This pruning established the retained feature-sets used in all subsequent analyses. As is evident from the table, the impact of pruning in this case was substantial, in some cases increasing prediction accuracy four- or six-fold.

The data also indicate that prediction of human similarity judgments from pruned GloVe embeddings can approach the noise ceiling. Bedny et al. (2019) provided a ceiling estimation: they obtained similarity ratings from a separate (third) group of sighted individuals, and evaluated the correlation between that group's ratings and those of the experimental sighted and blind groups. In some cases, such as for Light Emission verbs, high correlations were observed (0.91, 0.93 for sighted and blind) indicating high reliability of similarity ratings across different groups. In this case, supervised pruning achieved correlations of 0.85 and 0.88 respectively. Conversely, for Motion verbs, lower behavioral correlations were found (0.75, 0.72), with supervised pruning resulting in 0.73 and 0.68. This suggests that for specific semantic categories, prediction from pruned features approached the ceiling.

|  | CB Base | CB Retained | S Base | S Retained | CB F. | S F. |
|---|---|---|---|---|---|---|
| Perc. Amodal | 0.43 | 0.74 | 0.48 | 0.77 | 25 | 126 |
| Perc. Sight | 0.56 | 0.81 | 0.69 | 0.85 | 113 | 149 |
| Perc. Touch | 0.13 | 0.64 | 0.14 | 0.69 | 60 | 59 |
| Emission. Light | 0.64 | 0.88 | 0.49 | 0.85 | 55 | 94 |
| Emission. Animate Sound | 0.25 | 0.69 | 0.32 | 0.76 | 88 | 62 |
| Emission. Inanimate Sound | 0.10 | 0.60 | 0.16 | 0.64 | 32 | 32 |
| Motion | 0.36 | 0.68 | 0.35 | 0.73 | 57 | 32 |

Table 2: Comparison of prediction accuracy (Spearman's rho) when using full feature set (baseline) or retained features sets (retained). In this application, there was no test set.

Beyond this, we note the following. First, retaining more features did not necessarily produce better prediction accuracy: For Perception-Amodal verbs and Emission-Light verbs the number of retained features strongly differed between blind and sighted, but prediction accuracy (both baseline and retained) was similar. Second, pruning solutions that retained more features were not sparser. We compressed each retained embedding matrix and compared it to its original size (Ziv & Lempel, 1977). We found that across all solutions, the compression ratio was almost exactly 0.4 (range: $0.39 - 0.42$). This suggests that the different retained feature sets contained similar information, per dimension. And consequently, retained sets with larger numbers of features contain more information.

The Dice coefficient analysis (Figure 2a) indicated that the highest coefficients were found in the diagonal, which indexes cases of the same verb type, across sighted and blind. The highest value (0.89) was found for light-emission verbs (S_e_l, CB_e_l). Notably however, for Perception-Amodal, Motion, and Emission-InanimateSounds, the overlap was weaker. There was one case of strong overlap across verb types, seen in high Dice value for Perception-Amodal and Perception-Sight verbs, but only for sighted ($Dice = 0.42$). Importantly, for blind, this was not the case, with these verb types showing effectively no overlap ($Dice = 0.04$). This suggests that for sighted, Perception-Amodal verbs are associated with a stronger visual component. Many Dice coefficient were low, in the range of 0 to 0.05, with one reaching zero. Motion and Emission-InanimateSounds show particularly low values. To conclude, analyzing the overlap of features retained by pruning strongly suggests that different verb categories map onto different embedding dimensions.

Figure 2b shows there was no stable core of features that tended to be retained across the pruning solutions. For each of the 300 GloVe features, we computed how often it appeared across the 14 different retained sets. As the Figure indicates, no feature appeared in more than nine of the 14 solutions, and only 16 of the 300 appeared in 7-9 solutions. Thus, there is little support for a core set of features that is relevant for all verb categories. Interestingly, 19 of GloVe's 300 features were not retained in any of the 14 pruning applications. To understand the meanings they contain, we identified the top-20 words, from the dataset of Binder et al. (2016) for which these features produced the highest activation sum. The obtained words were: election, minister, banker, voter, arrested, politician, scream, party, airport, gunshot, diplomat, spoke, rally, businessman, glass, cathedral, driver, megaphone, commander, jury (with the first five scoring much higher than the rest). As can be appreciated, most of these are terms related to society, law and order, groups and social situations, which are indeed unrelated to the verb categories we used.

### 3.3 PROBING: INFORMATION IN RETAINED FEATURE SETS

As described in Section 2.3, from each set of pruned embeddings we learned a PLSR solution and used LOOCV to produce a prediction of all Binder's 534 words on 65 semantic attributes. These 65 attributes were condensed (via averaging) into 14 main semantic domains. As first indicator of quality of prediction, for each semantic dimension we evaluated the correlation between the 534 predictions and the ground-truth ratings. Prediction was often very good, and Figure 3a identifies several findings that speak to differences in the information selected by pruning from blind and sighted data.

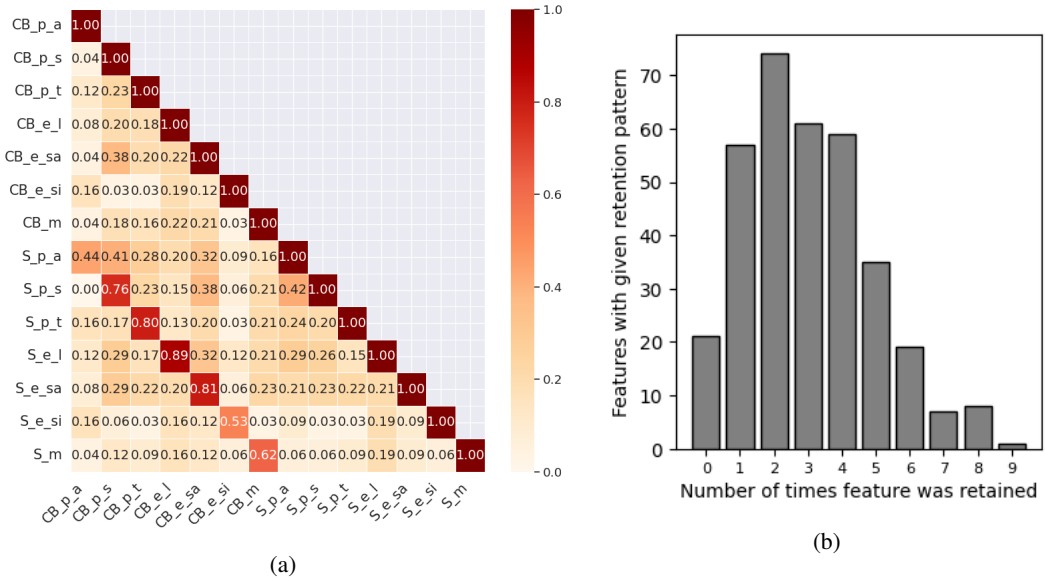

(a)

(b)

Figure 2: **Panel A**: Dice coefficients for different combinations of pruned sets. The size of the larger feature set was always matched to the size of the smaller one prior to computing the coefficient. p_a, p_s, p_t indicate Perception Action, Sight, and Touch verbs. e_l, e_sa, e_si indicate Emission of Light, Animate Sounds, and Inanimate Sounds. 'm' indicates Motion verbs. **Panel B**: The number of GloVe features ($sum = 300$) that appeared in zero or more retained sets. No feature appeared in more than nine of the 14 sets.

First, a methodological point is that accuracy of out-of-sample prediction of Binder data was not solely determined by the number of features in the retained set. For example, Perception-Touch verbs selected an almost identical number of retained features for blind and sighted (60, 59; see Table 2), but predictions based on features pruned for the blind provided better correlations with ground-truth data almost across the board, and particularly for the cognition domain. This means that similarity judgments for Perception-Touch verbs made by blind select for GloVe features that more strongly emphasize cognition-related information. Similarly, for sighted, Motion verbs and Emission-InanimateSounds verbs retained the same number of features ($n = 32$), with Motion better predicting the causal domain, and Emission-InanimateSounds better predicting the social and cognition domain. A second observation is that Perception-Sight verbs provided very good predictions, across the board, for both blind and sighted. Finally, the bottom row in the Figure (300d) presents prediction accuracy when using all 300 GloVe features as reference. It can be seen that while none of the retained feature-subsets reach these levels, the features retained by Perception-Sight verbs (for blind and sighted; S_p_s; CB_p_s) and those retained by Perception-Amodal verbs (for sighted) approximate this upper bound.

Table 3 shows which semantic domains were better predicted for sighted or blind, per verb category. Note that these results do not reflect over-fitting because they are based on predictions for out of sample data. For Perception-Amodal and Perception-Sight verbs, predictions for all semantic domain were more accurate when predicted from features retained for sighted. Controlling for multiple comparisons within each verb type ($p < .003, n = 14$ tests) we find that for Perception-Amodal verbs all differences were significant and for Perception-Sight, 10 of the 14.

For Perception-Touch verbs, five domains were better predicted for blind, but none were better predicted for sighted. This suggests that for blind, touch-related verbs are more strongly associated with other sensory functions, and also cognitive properties. Verbs of Emission-AnimateSounds also selected for much more information in blind, across both sensory and cognitive domains. We note that these verbs included words that are often indicative of particular mental states or event-types, e.g., moan, whimper, groan, growl, wail and it may be that for the blind these types of auditory signals serve as a more relevant way for communicating information than for sighted. This is in marked contrast to Emission-InanimateSounds (e.g., sizzle, buzz, creak, crackle) for which such

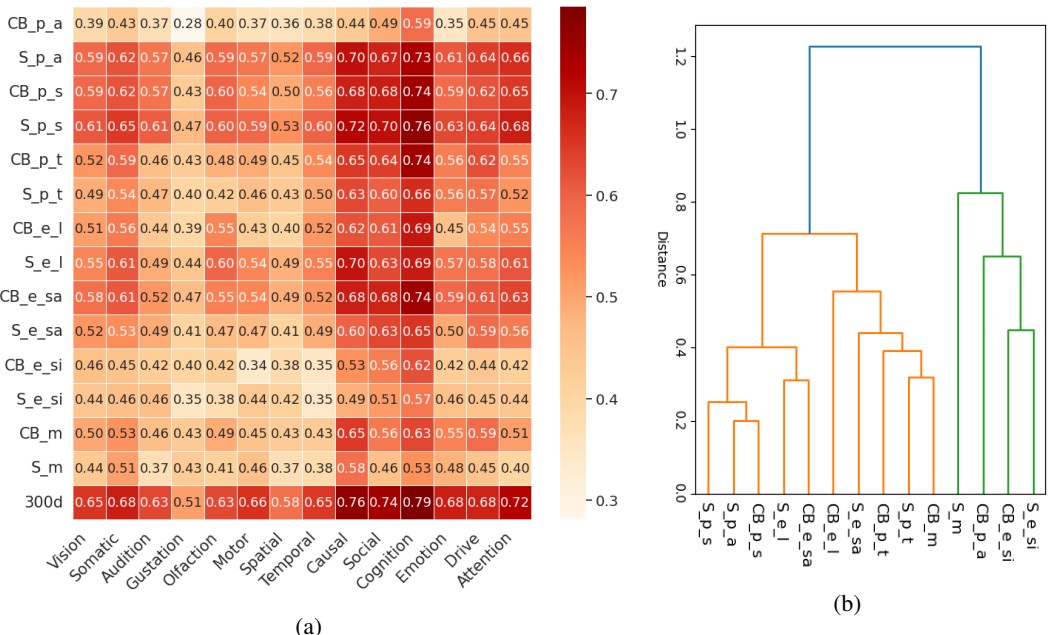

Figure 3: **Panel A**: Correlations between predicted and ground truth ratings when retained feature sets were used as regressors for predicting Binder's words. p_a, p_s, p_t indicate Perception Action, Sight, and Touch verbs. e_l, e_sa, e_si indicate Emission of Light, Animate Sounds, and Inanimate Sounds. 'm' indicates Motion verbs. The top row (300d) shows prediction accuracy when all 300 GloVe dimensions are used in the PLSR model. **Panel B**: Clustering of experimental conditions by prediction accuracy for the 65 Binder features. Only in the case of Emission-InanimateSounds are congenitally blind and sighted positioned in adjacent terminal leaves.

attributions are less relevant. Indeed, for the latter we found only one difference between the groups, where sighted better predicted the motor domain.

As indicated above, the PLSR method produced predictions for all 65 semantic attributes (see Appendix Figure 1). This provides an additional way to understand the information contained in the retained features: given the 14 pruned datasets (7 verb types for blind/sighted) one can apply clustering directly to the produced $14 \times 65$ prediction matrix. The clustering solution revealed that the terminal leaves rarely consisted of pairings of Blind and Sighted (see Figure 3b). Only for inanimate sounds were blind and sighted results positioned in adjacent terminal leaves (S_e_si, CB_e_si). These results suggest that the information contained in the retained embeddings, as assessed by the probing task, is not necessarily shared between congenitally blind and sighted for a given verb type.

## 4 DISCUSSION

By examining representations of different verb types from different domains, we find there are some domains for which the two populations share a common understanding, and some for which they strongly diverge. Our main contribution is identifying the semantic dimensions of verb meanings that are differently salient for blind and sighted.

Pruning presented good out of sample generalization which licensed the analysis of the retained feature sets and their use in the probing task. In most cases of out-of-sample testing, pruning matched or exceeded baseline accuracy level with 30-150 of GloVe's original 300 features. Dice analyses revealed that for certain verb categories, pruning from blind and sighted behavior retained similar features, but for others it retained different sets. We then used probing to evaluate whether these different feature sets described similar representational (latent) spaces. We used PLSR to predict 65 semantic properties for each left-out-word. This allowed us to evaluate the correlations between word-level predictions and ground-truth ratings (Figures 3a, 3b), as well as the raw distances (ab-

| Domain | P. Amodal | P. Sight | P. Touch | E. Light | E. A.S | E. I.A.S | Motion |
|---|---|---|---|---|---|---|---|
| Vision | **14.03** | **4.56** | **-3.75** | **5.10** | **-5.90** | -0.36 | **-4.13** |
| Somatic | **8.82** | **4.50** | -2.32 | 2.94 | **-5.33** | 0.93 | -0.57 |
| Audition | **9.67** | **3.17** | 1.12 | 2.75 | -2.79 | 0.77 | **-3.58** |
| Gustation | **8.86** | **4.09** | **-3.19** | **4.69** | **-6.29** | 1.63 | -2.72 |
| Olfaction | **9.13** | **3.56** | **-3.04** | **4.09** | **-5.30** | -0.44 | **-3.85** |
| Motor | **9.72** | **4.13** | -1.81 | **6.94** | **-4.23** | **4.90** | 0.20 |
| Spatial | **8.78** | **3.34** | -2.41 | **6.49** | **-3.64** | 2.92 | -2.20 |
| Temporal | **7.80** | **3.39** | **-3.18** | 3.07 | -1.56 | -0.57 | -0.89 |
| Causal | **9.00** | 2.51 | -1.96 | **4.45** | **-4.57** | -2.00 | -2.70 |
| Social | **9.42** | **4.17** | -2.39 | 1.53 | **-3.05** | -2.20 | **-4.76** |
| Cognition | **6.55** | 2.93 | **-4.13** | 1.12 | **-5.81** | -1.93 | **-3.46** |
| Emotion | **8.65** | 2.88 | 1.47 | **5.92** | **-4.89** | 0.68 | **-3.15** |
| Drive | **7.08** | 2.20 | -2.11 | **2.25** | -2.18 | 0.12 | **-5.04** |
| Attention | **7.69** | **3.00** | -0.68 | **2.84** | **-3.85** | 0.86 | **-3.23** |

Table 3: Differences in prediction accuracy of humanly derived semantic dimensions from retained GloVe features applied to the Binder et al. dataset. Each column corresponds to an analysis performed on features retained from similarity judgments of one verb type (7 in total). Numeric values indicate $T$ statistics from t-tests comparing prediction accuracy from sighted and blind. For each semantic dimension (rows), positive values indicate greater prediction accuracy for sighted, and negative values greater accuracy for blind. Values in bold are statistically significant (Bonferroni corrected for 14 contrasts in each column). P, Perception; E, Emission; E. A.S, Emission Animate Sounds; E. I.A.S., Emission Inanimate Sounds

solute error) between predictions and ground-truth ratings (Table 3). We find that the two groups produced very different prediction-accuracy profiles for Motion verbs and Perception-Amodal verbs. Specifically, the blind's representation of Motion verbs, as operationalized by the set of retained features, performed much better than sighted's in predicting higher-level cognitive concepts related to Causality, Cognition and Social information. Conversely, for Amodal verbs, sighted individuals drew on richer information than blind. This was confirmed the analysis of absolute prediction errors, which indicated that sighted individuals link more semantic contents (compared to blind) with Amodal verbs, whereas blind associate more information (compared to sighted) with Motion verbs.

The study has several methodological implications. First, supervised pruning allowed comparing representations of words for which pairwise similarity ratings were not obtained. Whereas RSA relates different Representational Dissimilarity Matrices (RDMs), it requires that the two RDMs are based on the same set of words (or objects more generally). Second, to our knowledge, this is the first application of a probing task to study interindividual differences in knowledge representation. We note that a probing task differs from canonical correlation and analysis and centered kernel alignment, methods commonly used to assess representation similarity (for review, see Klabunde et al., 2023). These methods quantify the relative similarity of representations within two matrices. We could have directly applied these techniques to the pruned word-embedding matrices (per verb type) to determine the similarity of information captured by sighted and blind, without using any probing task (i.e., without relying on data of Binder et al., 2016). However, the advantage of probing is that it not only indicates for which verb types do the representations of blind and sighted differ strongly or weakly, but also shows which semantic dimensions are more accurately predicted by information from one group or the other. It has been shown that differential predictions in probing tasks are mathematically linked to similarity metrics obtained through canonical correlation analysis (see Feng et al., 2020). An alternative methodology could have involved having both blind and sighted individuals rate verbs on experimenter-defined features (e.g., Kerr & Johnson, 1991). However a limitation of this approach is that there is no guarantee that all chosen features are cognitively relevant for representing the category. This means that, in the limit, blind and sighted may provide very different feature-ratings for a set of words (producing different representations in this feature space), but still judge the similarity of the words in the same manner because many of the features are less relevant.

A more general consideration relates to how (or whether) we should think of word-embeddings as analogs of human semantic knowledge. The current study suggests that word-embedding features do not capture a general representational space as a result of their training. Instead, relatively modular subspaces in the language model seem to match with different human concepts. These subspaces are described by limited subsets of features, which are highly sufficient for accounting for the geometry of particular human categories.

Because of the in-depth analysis applied to each set of verbs, our study relied on specific computational models and datasets, which produce several limitations. First, our ability to study differences in saliency of semantic dimensions was limited to those 65 dimensions made explicit in the probing task. Using additional datasets for the probing task could offer a more comprehensive view of differences between the blind and sighted groups. Second, we present results derived from a single variant of a pruning algorithm. Other algorithms may produce even more efficient pruning. Third, we used GloVe as a language model, pre-trained from a specific corpus. There are numerous other combinations of corpora and models trained on them, and they tend to produce different representational spaces (Lenci et al., 2022). However, GloVe is competitive with other word-embedding algorithms in predicting human similarity judgments, and in any case, differences between models appear minor (see Richie & Bhatia, 2021, their Figure 1).

In conclusion, we show that combining machine learning and behavioral methods allows studying inter-individual differences in word meaning for speakers of the same language. More specifically with respect to blindness, we find that it introduces important changes in both the amount of information associated with certain verb types, and the nature of the semantic dimensions related to verb meaning. Interestingly, this occurs for some verb categories but not others indicating that the organization of representation is domain or concept-specific.

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

# A  APPENDIX

## A.1  VERBS IN EACH CATEGORY FROM BEDNY ET AL. 2019

- PERCEPTION SIGHT verbs ($N = 14$): These were based on vision: gawk, gaze, glance, glimpse, leer, look, peek, peer, scan, see, spot, stare, view, watch.

- PERCEPTION TOUCH verbs ($N = 15$): These involved touching using the hands: caress, dab, feel, grip, nudge, pat, pet, pinch, prod, rub, scrape, stroke, tap, tickle, touch.

- PERCEPTION AMODAL verbs ($N = 15$): These involved cognitive activities related to knowledge acquisition: characterize, classify, discover, examine, identify, investigate, learn, note, notice, perceive, question, recognize, scrutinize, search, study.

- EMISSION (OF) LIGHT verbs ($N = 15$): Refer to events in the environment sensed through vision only: blaze, blink , flare, flash, flicker, gleam, glimmer, glint, glisten, glitter, glow, shimmer, shine, sparkle, twinkle. These are all associated with inanimate objects.

- EMISSION (OF) ANIMATE SOUNDS verbs ($N = 15$): Refer to events in the environment sensed through audition only and that are produced by humans or animals: bark, bellow, groan, growl, grumble, grunt, howl, moan, mutter, shout, squawk, wail, whimper, whisper, yelp.

- EMISSION (OF) INANIMATE SOUNDS verbs ($N = 15$): Refer to events in the environment sensed through audition only and that are produced by inanimate objects: beep, boom, buzz, chime, clang, clank, click, crackle, creak, crunch, gurgle, hiss, sizzle, squeak, twang.

- MANNER OF MOTION verbs ($N = 15$): refer to acts involving physically moving or changing position: bounce, float, glide, hobble, roll, saunter, scurry, skip, slither, spin, strut, trot, twirl, twist, waddle.

## A.2 FORMAL DESCRIPTION OF PRUNING AND PROBING

1.1 Matrix $M$ represents the embedding of $n$ words onto $d$ features within a language model ($M \in R^{n \times d}$). The word-pair similarity matrix, $Z$, is derived by computing the cosine similarity for each pair of rows in $M$ using $Z_{ij} = \frac{M_i \cdot M_j}{\|M_i\| \cdot \|M_j\|}$, with our focus being the upper triangle of matrix $Z$.

Our predictive target is the upper triangle of the human similarity judgment matrix $H$. We assume that a better approximation of matrix $H$ can be achieved by using a subset of columns in $M$, that is, by reducing the number of features ($d' < d$).

We therefore aim to identify matrix $M'$, which constitutes a submatrix of $M$, containing only selected columns. These selected columns are denoted as $S$, where $S \subseteq \{1, 2, \ldots, d\}$. $M' \in R^{n \times |S|}$ represents this submatrix. A similarity matrix $Z'$ derived from $M'$ is expressed as $Z'_{ij} = \frac{(M'_i) \cdot (M'_j)}{\|M'_i\| \cdot \|M'_j\|}$.

The primary objective of the pruning algorithm is to find the feature-subset $S^*$ that maximizes the correlation, measured using Spearman Rho, between the predicting matrix $Z'(S^*)$ and the target matrix $H$. To achieve this, the algorithm identifies a subset $S^*$ that maximizes the fit. The objective function is:

$$\arg\max \text{Spearman}\big(\text{vec}_{\text{upper}}(Z'(S)), \text{vec}_{\text{upper}}(H)\big).$$

As detailed in the main text, the pruning algorithm is supervised by similarity judgments from two distinct matrices representing Congenitally Blind and Sighted individuals. This yields two separate feature sets, denoted as $S^*_{\text{blind}}$ and $S^*_{\text{sighted}}$, which optimize the objective function. To evaluate whether these two sets of features define different representational spaces we use the logic of a probing task. Here, similarity between two representations is operationalized based on their performance in a downstream task. Two similar representations should perform similarly on a downstream prediction task.

Consider matrix $B$ as the representation of 534 target words in a 300-dimensional GloVe embedding space. The initial step involves creation of two submatrices: $B_{\text{blind}}$, utilizing exclusively $S^*_{\text{blind}}$, and $B_{\text{sighted}}$, utilizing exclusively $S^*_{\text{sighted}}$. (We follow here the terminology of Feng et al. (2020).) The downstream task is specified by matrix $Y$ ($Y \in R^{534 \times 65}$) where $Y_{(i,:)}$ represents the 65 human annotations for each word. This is the matrix to be predicted from the submatrices of $B$.

Let $p_W(B_{\text{blind}})$ and $p'_{W'}(B_{\text{sighted}})$ be two PLSR linear models applied to $B_{\text{blind}}$ and $B_{\text{sighted}}$ respectively, with $W$ and $W'$ representing the associated PLSR weights.

Using the least squares loss function $\ell(\hat{y}, y)$, the parameters $W$ and $W'$ are obtained by minimizing the following expressions, with $n = 534$:

$$\hat{W} = \arg\min_{W} \frac{1}{n} \sum_{i=1}^{n} \ell\big(p_W(B_{\text{blind}_i}), y_i\big)$$

$$\hat{W}' = \arg\min_{W'} \frac{1}{n} \sum_{i=1}^{n} \ell\big(p'_{W'}(B_{\text{sighted}_i}), y_i\big)$$

After obtaining $\hat{W}$ and $\hat{W}'$, the best fitting parameters for the two models (blind and sighted, respectively), it is possible to compare the two PLSR models $p_W(B_{\text{blind}})$ and $p'_{W'}(B_{\text{sighted}})$. This is done by quantifying by how strongly their predictions diverge, a property which Feng at al. refer to as *Transferred Discrepancy*. Two similar representations should produce very similar predictions. Note that this approach identifies similar representations even if the underlying vectors forming their bases exhibit substantial differences. That is, even if the columns selected in $S^*_{sighted}$ and $S^*_{blind}$ are different, as long as they reflect the same representation, they should produce similar predictions.

As introduced by Feng et al. (2020), the Transferred Discrepancy between the two representations is quantified by a divergence function $d()$, measuring the distance between the predictions derived from the two PLSR models (Equation 1).

$$TD(B_{\text{blind}}, B_{\text{sighted}}; Y) = \frac{1}{n} \sum_{i=1}^{n} d(p_W(B_{\text{blind}_i}), p'_{W'}(B_{\text{sighted}_i})) \tag{1}$$

We follow this approach with two minor modifications. First, rather than defining $TD$ based on a single (point) mean statistic, we compare the two-models' predictions using a test of differences between distributions, considering the entire set of predictions made by each model (534 predictions in each case). To this end we use paired-sample T-tests with words $n = 534$ as unit of analysis. Second, we define the divergence function $d()$ as the absolute distances between the predicted value of each model and the true values. These absolute magnitudes are compared in order to understand whether one produces more accurate predictions than the other. The results are reported in Table 3.

---

**Algorithm 1** Pruning

---

1. Inputs: $SM_{HM}$: Similarity Matrix of human similarity judgments. $SM_{DNN}$: Similarity Matrix of similarity estimations derived from the DNN by computing the cosine similarity between the embeddings of two words.
2. Compute baseline $\rho(SM_{HM}, SM_{DNN})$, from the full set of features
3. **Rank features**
   - Remove the first feature from all the original embeddings and compute the corresponding similarity matrix $SM_{DNNRED}$;
   - Compute the difference $D = \rho(SM_{HM}, SM_{DNN}) - \rho(SM_{HM}, SM_{DNNRED})$. $\rho$ is Spearman's rank correlation. Higher positive values for $D$ indicate that the removed feature was important;
   - Repeat the step above for all the possible $N-1$ feature subsets (where $N = 4096$);
   - Rank the features based on $D$.
4. **Construct pruned embeddings**
   - Starting from an empty set of features, construct pruned embeddings by iteratively reinserting one feature at a time, in descending order of importance;
   - Compute Spearman $\rho$ after each feature reinsertion and store the values in the array $a$;
   - Compute the maximum of $a$. Its position (index) within the array delimits the set of features to be included in the pruned embeddings.

---

A.3 PREDICTION ACCURACY FROM PRUNING SOLUTIONS

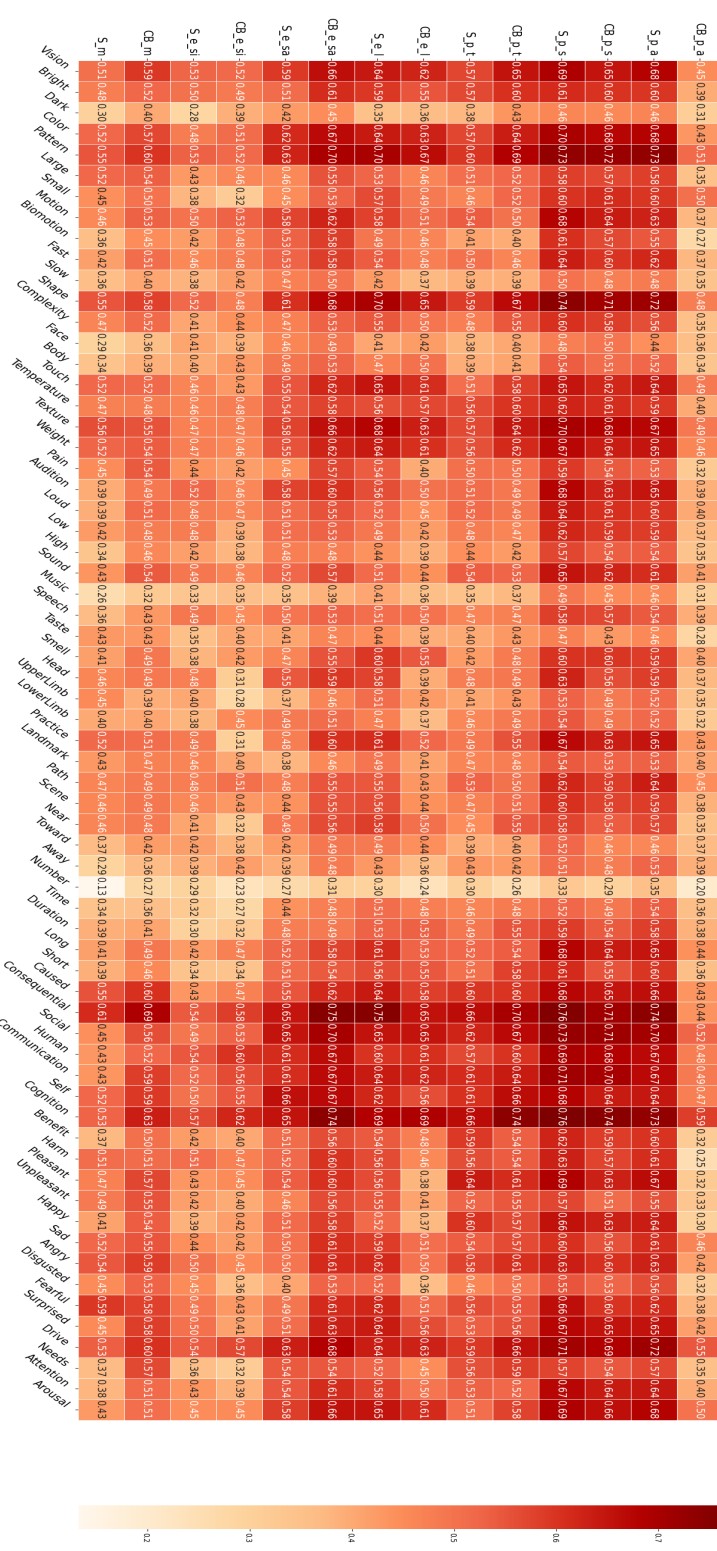

Figure 1: Prediction accuracy for each of the 65 features in Binder et al., from each of the 14 retained features sets obtained via pruning

