# OpenReview forum: "Identifying and Interpreting Non-Aligned Human Conceptual Representations using Language Modeling"
_ICLR.cc/2024/Workshop/Re-Align — ICLR 2024 Workshop Re-Align Poster_

### Official Review · Reviewer_T4cm · 2024-02-21
**Interesting method and promising results in a unique case study, displaying differences in representations/conceptual basis and explanations of the differences.**

**Rating:** 3
**Fit:** 3
**Confidence:** 2

**Workshop Review:**

This paper introduces a supervised representational alignment method using feature-pruning to a language model to optimize prediction accuracy of human similarity judgments from word embeddings. The method determines whether two groups of individuals share the same basis of a category and explains how they differ. They present the effectiveness of the method in application to sighted individuals and congenitally blind, demonstrating that congenital blindness induces conceptual reorganization and the associated semantic shifts can be identified. The paper uses language model features which were useful in predictions of judgments made by the congenitally blind vs. sighted individuals followed by linear probing to interpret the latent semantics of the subset of features for each group, which are mapped to a number of interpretable semantic domains. The experiments are well done, covering several diverse semantic domains, from verbs related to sight, to touch, to amodal verbs.

This is a very interesting approach, the experiments are well constructed, and the results properly support the claims of the paper. I strongly recommend the paper for presentation at the workshop.

Minor notes:
-Figure 1 displays a bit blurry
-I'd appreciate some clarification/extra detail in section 2.3 about what's procedurally done when the authors say they condense the 534 x 65 matrix into 534 x 14 (the number of feature sets)
-A clear explanation on the author's thoughts for the limitations of the approach, especially about the degree to which we can claim understanding/explanation of the different sets of important features, would be very valuable.

**Reason For Not Giving Higher Score:**

N/A

**Reason For Not Giving Lower Score:**

Very strong methodologically, applies existing tools in an interesting case study, positive finding in this domain is well supported, interesting, and would be of interest to the target of the workshop.

**Reviewer Domain:**

cognitive science

---

### Official Review · Reviewer_iGmr · 2024-02-22
**Interesting method that uses the GloVe language model to study a challenging question of how the conceptual representations of the blind and sighted differ**

**Rating:** 2
**Fit:** 2
**Confidence:** 1

**Workshop Review:**

Summary:
- The paper introduces a representational alignment method (Section 1.2) that involves Partial Least Squares Regression (PLSR) to interpret how the conceptual representations of the blind and sighted are similar and different. They identify the semantic dimensions of verb meanings that are differently salient for the blind and sighted.

Strengths:
1. Interesting method that uses the GloVe language model to study a challenging question of how the conceptual representations of the blind and sighted differ.

Weaknesses:
1. The method involves multiple steps where errors may accumulate, e.g., language models like GloVe do not capture all relevant semantic concepts, supervised pruning has imperfect accuracy, human similarity judgments are on a linear and one-dimensional scale of 1 to 7, and the probing method has imperfect accuracy. I think this introduces a greater accumulation of error than if the blind and sighted directly rated the same set of words on the 65 semantic dimensions from the paper they cited (Binder et al., 2016).

Minor notes:
1. GloVe is a pretty good model but is somewhat old (2014) and has limitations compared to newer language models. One issue is that the expressiveness and semantic dimensions identified will be limited to the conceptual representations that are captured by GloVe.

**Reason For Not Giving Higher Score:**

N/A

**Reason For Not Giving Lower Score:**

N/A

**Reviewer Domain:**

machine learning

---

> ### Author Response · Authors · 2024-03-11
> **Regarding error, error propagation, conclusions from direct-rating procedure, and GloVe**
>
> The reviewer expresses concern that the sequential application of pruning and probing might introduce cumulative inaccuracies because of language-model limitations, imperfect pruning and imperfect probing.
>
> We thank the reviewer for these constructive and important points, and we have consequently made several revisions, which appear in the Camera Ready version.
>
> ** Regarding effectiveness of pruning: In the results we now report data suggesting that pruned-GloVe embeddings can achieve performance approaching the expected noise ceiling in predicting human similarity judgments. Regarding noise ceiling, Bedny et al. (2019) provide an effective ceiling estimation using human similarity ratings from a separate reference group of sighted individuals. Specifically, they had another group of sighted individuals make similarity judgments for the materials and evaluated the second-order-isomorphism (Pearson’s Rho) between the ratings of that reference group and those of the experimental sighted and blind groups (see their Figure 3A). These correlations indicate a prediction ceiling as they capture the precision of the measurement itself. In some cases, such as Light Emission verbs, behavioral correlations were high (0.91, 0.93 for sighted and blind) and supervised pruning reached 0.85 and 0.88 respectively. In other cases, such as motion verbs, behavioral correlations were lower (0.75, 0.72) and supervised pruning produced 0.73 and 0.68.  This suggests that supervised pruning can achieve high sensitivity for specific semantic categories.
>
> ** Regarding probing, while noise ceiling estimations were unavailable, we find good out-of-sample prediction accuracy when using all 300 dimensions. For example, for some words, the correlation between the human-rated values on the 65 dimensions and the ones predicted using leave-one-out CV exceeded R=0.93. The average correlation values across 534 words was R=0.72. This indicates that probing can achieve good, if not perfect accuracy. We do not report this in the manuscript, because in the original publication reporting these data (Binder et al., 2016) there is no data directly informative about the noise ceiling and so a reference value is unavailable.
>
> ** Regarding direct-rating procedures of the type suggested in the comment: while directly asking blind and sighted individuals to rate verbs on experimenter-defined features offers an alternative way to study representation, it has limitations. In this case, there is no guarantee that all chosen features are indeed cognitively relevant for the representation of the category. This means that, in the limit, blind and sighted may provide very different feature-ratings for a set of words (producing different representations when those words are positioned in *this* feature space), but still judge the similarity of the words in the same way because many of the features rated are less relevant.  We have added a clarification regarding this point in the discussion.
>
> ** Finally, regarding GloVe: we agree, and it may be that FastText has greater potential in predicting human similarity (Lenci et al, 2022; https://doi.org/10.1007/s10579-021-09575-z , see their Figure 3).  However, Lenci et al. also show that no single language model performs best across different similarity-judgment datasets (see their Table 4), and so there is no a-priori way to select a most effective model. The use of contextual embeddings is further complicated by how type embeddings are to be computed from contextualized context vectors. Lenci et al. average BERT representations, and find that the embeddings from static models typically outperform these in predicting human similarity (see their Table 5).

---

### Official Review · Reviewer_jqnq · 2024-02-24

**Rating:** 3
**Fit:** 3
**Confidence:** 2

**Workshop Review:**

This paper presents results of an empirical analysis of word meanings in humans. The "alignment" discussed here is between two groups of individuals -- blind people and sighted people.

Strengths include a thorough, in-depth and well-explained analysis. Experiments are well-motivated and the types of words used are also meaningful.
Weaknesses include limited discussion of the implications for human-AI interactions, alignment, etc.

The paper would benefit from tying the motivations and results back to themes in AI research or some kind of human-AI alignment. I can think of some already, and the results (although technically presenting human-human alignment results) are clearly relevant to researchers of representational alignment for AI as well ... but this connection may not be immediately obvious for all workshop attendees.

**Reason For Not Giving Higher Score:**

N/A

**Reason For Not Giving Lower Score:**

The paper was sound in all respects -- motivation, background research, experiment design, data analysis, and discussion of results/meaningful impact. All three of the relevant communities at this workshop would be interested in the results.

**Reviewer Domain:**

machine learning

---

### Decision · Program_Chairs · 2024-03-02

Accept (Poster)